# Does the Composition of the Gut Bacteriome Change during the Growth of Tuna?

**DOI:** 10.3390/microorganisms9061157

**Published:** 2021-05-27

**Authors:** Elsa Gadoin, Lucile Durand, Aurélie Guillou, Sandrine Crochemore, Thierry Bouvier, Emmanuelle Roque d’Orbcastel, Laurent Dagorn, Jean-Christophe Auguet, Antoinette Adingra, Christelle Desnues, Yvan Bettarel

**Affiliations:** 1MARBEC, University Montpellier, CNRS, Ifremer, IRD, 34095 Montpellier, France; elsa.gadoin@umontpellier.fr (E.G.); durandlucile.ld@gmail.com (L.D.); aurelie.guillou@ird.fr (A.G.); Sandrine.CROCHEMORE@cnrs.fr (S.C.); thierry.bouvier@cnrs.fr (T.B.); emmanuelle.roque@ifremer.fr (E.R.d.); laurent.dagorn@ird.fr (L.D.); Jean-christophe.AUGUET@cnrs.fr (J.-C.A.); 2Centre de Recherches Océanologiques, Abidjan, Côte d’Ivoire; adingraantoinette_ama@yahoo.fr; 3Mediterranean Insitute of Oceanodraphy (MIO), 13009 Marseille, France; Christelle.DESNUES@univ-amu.fr

**Keywords:** tuna, microbiome, enteric bacteria, fish, barcoding, gut

## Abstract

In recent years, a growing number of studies sought to examine the composition and the determinants of the gut microflora in marine animals, including fish. For tropical tuna, which are among the most consumed fish worldwide, there is scarce information on their enteric bacterial communities and how they evolve during fish growth. In this study, we used metabarcoding of the 16S rDNA gene to (1) describe the diversity and composition of the gut bacteriome in the three most fished tuna species (skipjack, yellowfin and bigeye), and (2) to examine its intra-specific variability from juveniles to larger adults. Although there was a remarkable convergence of taxonomic richness and bacterial composition between yellowfin and bigeyes tuna, the gut bacteriome of skipjack tuna was distinct from the other two species. Throughout fish growth, the enteric bacteriome of yellowfin and bigeyes also showed significant modifications, while that of skipjack tuna remained relatively homogeneous. Finally, our results suggest that the gut bacteriome of marine fish may not always be subject to structural modifications during their growth, especially in species that maintain a steady feeding behavior during their lifetime.

## 1. Introduction

In recent years, the number of microbiome studies on marine organisms was increasing, particularly for corals and fish, about which we now know that their microbial associates play a considerable role in their health and fitness [1,2,3,4]. Among the different biological compartments that harbor microorganisms, the digestive tract is certainly the one that has received the greatest deal of attention; mostly because enteric bacteria are involved in a wide range of important functions for the host, including digestion, production of useful metabolites, protection against pathogens, promotion of the immune system, behavior, to name a few [5,6,7,8]. Previous studies have shown that for humans, the gut microbiome of fish include, from the larval stage onwards, a wide range of taxa mostly dependent on several factors such as life stage [9], sex [10], phylogeny [11], trophic level [12], diet [7], season [13], habitat [14] and captive-state [10].

At various growth stages, the life traits of fish may evolve and could therefore result in major changes in the composition of enteric bacterial communities [6,15,16]. This is the case, for example, in wild migratory species that undergo several ontogenetic transformations during their development and need to adapt their metabolism and diet to major environmental changes, such as the transition from fresh water to salt water [17]. A handful of studies revealed that the gut microbiome changes throughout fish growth, as shown in the farmed olive flounders *Paralichthys olivaceus* [18] and Chinook salmon *Oncorhynchus tshawytsha* [19]. However, these studies have been mostly conducted on experimental reared fish, hence it is timely to evaluate whether such patterns also occur in wild fish, especially for species that represent an important food source for humans.

Tunas are teleostean species living in tropical and temperate waters. They are top predators playing a fundamental role in the marine trophic food chain and ecosystem resilience. They are among the most fished species in the world, exploited in all oceans by both industrial and small-scale fisheries [20].

Despite their ecological importance, but also in terms of ecosystem services provided by tunas, little is currently known about the composition, diversity and role of their gut bacteriome, including its possible changes during fish growth. Knowing that the ontogenic changes typical of certain tuna species allow them to access additional sources of food during their growth [21], it is possible that the diversification of their diet with age could result in substantial modifications of their gut bacteriome.

The objective of this study was (i) to describe the composition of the gut bacteriome of the three most fished tuna species (skipjack-*Katsuwonus pelamis,* yellowfin-*Thunnus albacares* and bigeye-*Thunnus obesus*), and (ii) to investigate whether the fish size (as a proxy of their development stage) is a decisive factor in explaining the structure of this enteric community.

## 2. Materials and Methods

### 2.1. Sampling

The protocol consisted of collecting 18 individuals of each of the three tuna species targeted by the large-scale purse seine fleet. All individuals were caught between May and December 2019 (Figure 1) in the Eastern Atlantic Ocean (Gulf of Guinea and off the coast of Senegal) and sampled by the Exploited Tropical Pelagic Ecosystem Observatory (IRD, Ob7), as part of the multiannual European fishery data collection framework (DCF, financed by the European Maritime and Fisheries Fund, Article 77). Once the fish had been caught, they were stored onboard in chilled brine to lower their temperature to around −15 °C.

Each fish was weighed (whole weight, in kg) and lengthed (at fork, in cm). Size ratio (SR) between the largest and smallest individuals was calculated for each species, following the equation: SR = minimal size/maximal size. For each species, the individuals were grouped into three equivalent categories (in terms of number), named Small, Medium and Large, and corresponding, respectively, to the 6 smallest, medium and largest fish per species (Table 1). Considering the size at 50% sexual maturity of each species [22], individuals from the Small group can be considered as being mainly juvenile or sub-adult fish, and individuals from the Medium group as sub-adults or young adults, while the Large group for each species can be considered to be composed of adults. The complete set included male, female and immature fishes but no significant difference in size or weight were observed between these three categories (*p* > 0.05, Kruskal-Wallis test).

### 2.2. Extraction of the Gut Bacteriome

After landing, the 54 frozen fish were transferred to the Laboratory of Microbiology of the Centre de Recherches Océanologiques (CRO) of Abidjan, where they were thawed and dissected. The time length between catch and dissection was approximately 40 days and it was comparable for each sampled fish. Briefly, the gastrointestinal tract was extracted from each individual and cut from below the stomach to 2 cm before the rectum (to avoid potential chilled brine intrusion), using sterile tools. Each gut was squeezed to expel the contents (from 3 to 15 mL) on a sterile surface, and the contents were homogenized before sampling [23] and conserved at −80 °C until the DNA extraction.

### 2.3. DNA Extraction, Amplification and Sequencing

Bacterial DNA was extracted from 250 ± 0.5 mg of gut (*n* = 54). All extractions were performed with the PowerSoil DNA Isolation Kit (Qiagen^®^, Hilden, Germany) following the manufacturer’s instructions. DNA quality and quantity were assessed by spectrophotometry (NanoDrop^®^, Wilmington, DE, USA). The V3-V4 region of the 16S rDNA gene was amplified using universal bacterial primers modified for Illumina sequencing: 343F (5’- ACGGRAGGCAGCAG) [24] and 784R (5’- TACCAGGGTATCTAATCCT) [25]. The reaction mixture consisted of 12.5 μL of 2X Phusion Mix (New England Biolabs^®^, Ipswich, MA, USA), 1 μL of each primer at 10 μM (Eurofin^®^, Luxembourg), 10 ng of DNA template and enough molecular-grade H_2_O (Qiagen^®^) to reach a final volume of 25 μL. All samples were amplified in triplicate and pooled to avoid PCR bias in the taxonomic diversity of the community [26]. Successfully amplified samples (*n* = 54) were sequenced on the Illumina platform (Genoscreen^®^, Lille, France) using 2 × 250 bp MiSeq chemistry.

### 2.4. Treatment and Analysis of the Bacterial Sequences 

In total 1,823,118 reads were obtained and were processed on RStudio (R v. 3.5.3) with the DADA2 pipeline (v. 1.10.1) [27], following the authors’ tutorial (https://benjjneb.github.io/dada2/tutorial.html, accessed on 27 February 2019). The quality of the forward and reverse reads was assessed prior to the removing of adaptors and primers, based on their length. Reads were then filtered, trimmed and merged into 583,716 amplicon sequence variants (ASV), which have a greater resolution than the operational taxonomic unit (OTU) [27]. The chimeras were removed and the paired sequences were aligned with the SILVA 123 taxonomic database [28] to access their taxonomy. To compensate for varying sequencing efficiency, analyses were performed on a random subsample of 8979 sequences per sample, which corresponded to the sample with the smaller number of sequences after trimming and quality processing. Final taxonomic and ASVs tables were associated with the morphometric data using the *phyloseq* package [29]. Finally, relative abundances of ASVs in each sample were calculated by the *phyloseq* package and ASVs corresponding to archea, non-prokaryotes, chloroplasts and mitochondria were deleted.

To assess the alpha diversity, the Shannon diversity index was calculated by *phyloseq* for each sample. The composition of bacterial communities was represented at the order level, based on the relative abundance of ASVs in each sample and performed with *phyloseq* and *ggplot* packages. Finally, using the microbiome package [30], the *Core Microbiota*, defined here as all ASVs (relative abundance ≥ 1%) shared by at least 50% of the gut samples, was determined for the three size groups of the three tuna species.

### 2.5. Statistical Analysis

All the statistical analyses were carried out using RStudio (R v. 3.5.3). For each species, possible relationship between the alpha diversity index (Shannon) and the size were tested by linear regression, while Kruskal-Wallis and Wilcoxon non-parametric tests were used to observe the variation of the alpha diversity between the three species. Finally, the effects of the species then of the size on the bacterial composition was determined by one-factor PERMANOVA with 999 permutations of the Bray-Curtis matrix using the “adonis” function of the *vegan* package [31].

## 3. Results

### 3.1. Fish Morphometric Traits

For the three target species, the size ratio between the smallest and largest individuals was comparable, reaching 2.2, 2.5 and 2.3 for skipjack, yellowfin and bigeye tunas, respectively. Skipjack were significantly smaller (30.5 to 67.5 cm) than the yellowfin (66.1 to 164.3 cm) and the bigeye tuna (71.4 to 166.8 cm) (*p* < 0.001, Kruskal-Wallis test).

### 3.2. Alpha Diversity

The Shannon diversity index varied considerably for each of the three species of tuna (Figure 2A). On average, the index for the skipjack tuna was significantly lower than for the two other species (Kruskall-Wallis, *p* < 0.05). However, for all the three species, the Shannon index did not vary significantly along the size gradient (R2_Skipjack_ = 0.13, *p* = 0.07; R2_Yellowfin_ = 0.11, *p* = 0.09; R2_Bigeye_ = 0.13, *p* = 0.07) (Figure 2B).

### 3.3. Composition and Beta Diversity

Forty-five orders of bacteria belonging to 15 different classes were identified (Figure 3). *Actinobacteria*, *Alphaproteobacteria* and *Gammaproteobacteria* were the most represented classes, regardless of the species and size of tuna. Generally, there was a significant species-effect on the enteric bacteriome of tuna, which was particularly marked for the gut skipjack individuals, which was dominated by *Mycoplasmatales* (*Mollicutes*) (Figure 3).

Regarding the variability in the gut microflora during fish growth, two different patterns were observed: while the composition of enteric bacterial communities was significantly affected by fish size for yellowfin and bigeyes tunas, that of skipjack remained relatively stable across the range of fish sizes (Table 2).

### 3.4. The Core Gut Microbiome

During their growth, each species hosted taxa that were common and specific to one or more size groups. Twelve genera of bacteria were found in the gut of all the three different tuna species (Table 3), some of them such as *Carnobacterium* sp., *Oceanisphaera* spp., *Pseudomonas* spp. and *Psychrobacter* spp. were even found in all the three size groups (small, medium and large). Each tuna species had also its own specific taxa, such as *Mycoplasma* sp. in skipjack, in all size categories, and to a lesser extent, *Corynebacterium* sp. and *Rahnella* spp. Yellowfin tuna hosted specific bacterial taxa mainly belonging to the *Alphaproteobacteria*, *Clostridia*, *Fusobacteriia* and *Gammaproteobacteria* classes, the latter being the most relevant, with taxa such as *Citenobacter* sp., *Aeromonas* sp., *Massilia* sp., and *Photobacterium* sp. Taxa such as *Microbacterium* sp., *Labrenzia* sp. *Vibrionimonas magnilacihabitans* and *Escherischia/Shigella* were specific to the bigeye’s gut.

Some taxa were characteristic of a size group, this being very marked for yellowfin which had the highest number of size-specific taxa. Taxa such as *Acinetobacter* sp. and *Aeromonas* sp. were specific to medium sized yellowfin and others such as *Photobacterium angustum* and *Photobacterium leiognathi,* both reported as histamine producing bacteria, were found only in large yellowfin (Table 3).

## 4. Discussion

In the three tuna species, the gut bacteriome was dominated by four main phyla: *Proteobacteria, Firmicutes*, *Bacteroidetes* and *Actinobactera,* which accounted for up to 95% of the bacteria identified in the gut. Such phyla are typically found in the intestinal microflora of marine and freshwater fish [6,32]. However, the diversity of the 16S rDNA gene sequences showed considerable differences between the enteric bacterial communities of the three species (Figure 2 and Figure 3). Skipjack were distinguished from yellowfin and bigeye tunas by their particularly low species richness and a strong presence of *Firmicutes* (*Mollicutes*, *Mycoplasma* sp.). A low gut microbial diversity, associated with the dominance of *Mycoplasma* spp. was also reported for Atlantic salmon (*Salmo salar*) [17]. So far, within the marine environment, *Firmicutes* were thought to be the dominant phylum in herbivorous species [11,33], probably as they facilitate the digestion of cellulose. However, this metabolic function is not vital in tunas and Salmonids, which are carnivorous fish. By comparison with yellowfin and bigeye, skipjack tunas are also more subjected to risk of overheating as they grow mostly because of their lower capacity of thermoregulation [34]. The occurrence of temperature-induced changes in the gut microbiome is a well-known phenomenon in vertebrates and usually results in a disruption of the alpha-diversity towards a decrease in richness [35]. Identical microbiome responses to thermal shifts in phylogenetically distant animal taxa suggest the existence of a conserved mechanism, which could also apply in tuna. All together, these may explain the observed lowest alpha diversity in skipjack.

Although our results indicate a clear species effect on the enteric bacteriome of tuna (Figure 3), there is nevertheless a strong convergence in the structure of the gut microbiota between yellowfin and bigeye while skipjack (both belonging to the same genus *Thunnus*)*,* whereas skipjack tuna (genus *Katsuwonus*) are in a separate branch). The composition of the gut bacteriome in tropical tuna could thus depend mainly on evolutive considerations [36]. Indeed, phylogenetically close fish usually host a similar bacterial flora [37]. However, the formation of the bacterial communities in the gastro-intestinal tract of fish is a complex process affected by other exogenic and endogenic factors such as the diet, the life style and the environment [6,37,38].

### 4.1. Microbial Changes during Fish Growth

For all the three species, the taxonomic structural diversity of enteric bacteria did not vary significantly with size (Figure 2). This is contradictory with recent reports showing a reduction of the alpha diversity with age, and therefore, with size, for Atlantic Salmon, olive flounders and zebra fish [17,18,39,40]. A reduction in diversity is usually associated with a diet that encouraged the growth of generalist bacteria or that included chemical compounds inhibiting certain more specialist bacteria [41]. Conversely, in terrestrial vertebrates (mammals, birds and reptiles) a positive correlation was observed between the enteric microbial diversity and the mass of the animal, independent of the age, phylogeny, diet or the structure of the digestive tract [42]. Overall, the changes in the structural diversity of enteric bacteria during bodily growth does not seem to follow a unidirectional pathway within the animal world and more studies are needed to identify the factors that govern this particular diversity.

The examination of the core microbiome demonstrated that changes in the proportions of the main taxa during growth were specific to each tuna species (Table 3). Although a core bacterial community was found across the three tuna species (comprising four major genera: *Carnobacterium, Oceanisphaera, Pseudomonas* and *Psychrobacter)*, other genera were specific to one or more size groups for each species. In particular, for yellowfin, *Acinetobacter* and *Aeromonas* were found in medium sized fish while *Cutibacterium*, *Lactococcus*, *Gottschalkia* and *Photobacterium* spp. were found only in large individuals. These taxa might, therefore, play a specific role in the late development stage of yellowfin.

### 4.2. Dietary Changes during Growth

The most striking result of our study was a drastic change in the composition of the gut microbiome of yellowfin and bigeye tuna during their growth, which was not observed for skipjack (Table 2, Figure 3). The fish size could have an effect on the selection of prey and some studies demonstrated that the proportion of fish in the tuna diet increases as the size of the tuna increases [43,44]. A vertical distribution of tuna species in the water column has been long reported, with the larger fish (bigeye and yellowfin) able to stay in deeper water than the smaller (skipjack), which gives them access to different types of prey [21]. Such modifications in the diet could be related to changes in nutritional needs depending on the development stage of the tuna. Indeed, ontogenic changes in yellowfin and bigeye tuna are generally observed when they reach about 45–50 cm, enabling them to dive into colder, deeper water [45], which would radically change their diet, and promote its diversification throughout growth. Quite the opposite, skipjack are physiologically enabled to reach these deeper waters with potential new preys and therefore remain in surface waters at all life stage, which could explain the homogeneity of their gut microbiota throughout growth. Conversely, although there was no yellowfin or bigeye smaller than 65 cm in our sampling, we suspect that the significant differences observed between the various growth stages was due to their ability to diversify their diet with age, going further and further to seek food.

### 4.3. Commensal and Potential Pathogenic Bacteria

In our study, the main bacterial genera forming the core microbiome of the tuna species included commensal and potential pathogenic taxa, some of them being common to all the three target species (Table 3). Of the commensal taxa, some *Carnobacterium* species, for example, are known to inhibit the action of certain fish pathogens [46,47]. A similar antagonistic activity of several *Pseudomonas* bacterial strains has also been reported [48]. Many *Kocuria* species are commensal bacteria found on mammals and have been isolated from the gut of rainbow trout (*Oncorhynchus mykiss*) and the Atlantic cod (*Ghadus morhua*) [49]. However, several species of *Carnobacterium* and *Kocuria* are also known to be pathogenic [49,50], like *Escherichia* and *Shigella,* which are enteric human pathogens able to establish in the gut of several fish in certain conditions. This has occurred, for example, in trout after consuming infected food and in Nile tilapia when the surrounding water had been contaminated [51,52]. Other *Sporocarcina* species are bacteria that spoil seafood and refrigerated meat [53]. Some species found in the tuna gut microbiome are also known to be histamine-producing bacteria (HPB) [54]. The histamine is produced by these bacteria from a precursor (histidine) by a bacterial enzyme (histidine decarboxylase) and is main global cause of food-poisoning from consuming fish [55]. HPB are often found in tuna when the cold chain is broken during landing, processing and handling fresh tuna [56,57]. In this study, two *Photobacterium* species (*Photobacterium angustum* and *Photobacterium leiognathi*) were found in the gut microbiome of large yellowfin. Other species such as *Photobacterium damselae*, *Photobacterium phosphoreum* and *Pseudomonas fluorescens* were found in some samples of all three tuna species. The relative abundances were low (<2%) and there was no clear relationship between the size of the tuna and the presence of these HPBs.

All these confirm that the fish gut typically hosts a complex and highly diversified bacterial community composed of a balance of commensal and pathogenic taxa, which contribute to the functioning of the gut and help to maintain the health of the host organism.

## 5. Conclusions

Our results revealed that the composition of the enteric microflora showed contrasting patterns between skipjack on one side, and yellowfin and bigeye tuna on the other side. Beside phylogeny, several other endogenic factors could explain the microbial differences and similarities between species, including the size which emerged as a major determinant of gut bacteriome in tuna. If significant changes in the intestinal microflora have been observed during the growth of yellowfin and bigeye tuna, the case of skipjack, by contrast, is interesting because of the relative stability of its microbiota and its unique composition. Overall, our study suggests that strong structural (and presumably functional) microbiological differences exist between species within the same family of fish, probably linked to their differential ability to grow in size, improve their mobility for foraging, and thus promote diet diversification.

## Figures and Tables

**Figure 1 microorganisms-09-01157-f001:**
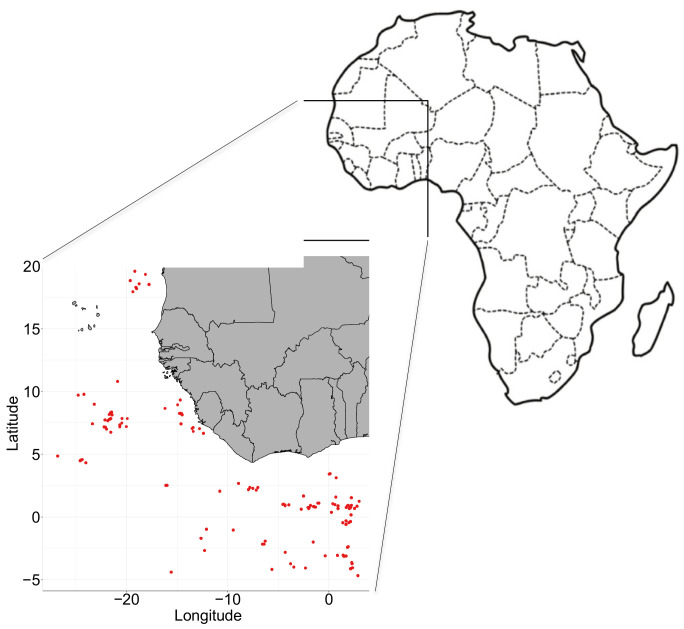
Fishing positions in the east Atlantic Ocean. Colored circles correspond to tuna caught from shoals near fish aggregating devices or from free swimming schools.

**Figure 2 microorganisms-09-01157-f002:**
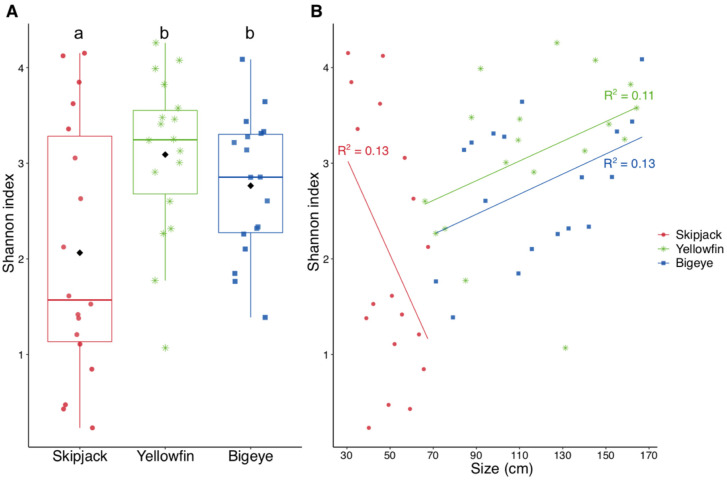
Representation of *alpha* taxonomic diversity (Shannon index) of enteric bacterial communities (ASVs) in the three tuna species (**A**), and according to the size of individuals (**B**). Boxplots represent the distribution of the alpha taxonomic diversity within each species. Different letters indicate significant differences (KW, *p* < 0.05) between groups within each square. * Significant correlation between the Shannon index and fish size (Pearson, *p* < 0.05).

**Figure 3 microorganisms-09-01157-f003:**
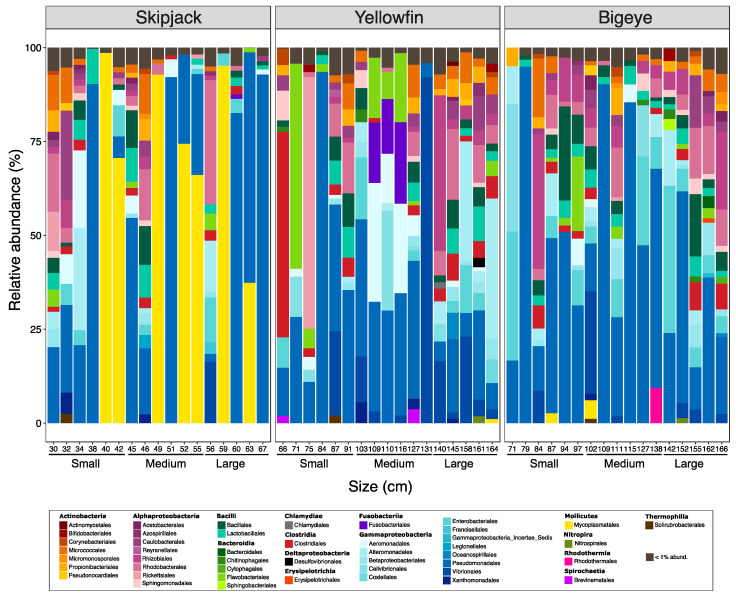
Relative abundances of the main bacterial classes in the gut of skipjack, yellowfin and bigeye tunas, in their size group (small/medium/large). Each bar corresponds to an individual fish. Bacterial classes showing a relative abundance lower than 1% were designated “<1% abund”.

**Table 1 microorganisms-09-01157-t001:** Main morphometric and sexual traits of the fish sampled in the three size categories (small/medium/large). M, male; F, female; I, Immature.

	SKIPJACK			YELLOWFIN			BIGEYE		
Size Class	Size (cm)	Weight (kg)	Sex	Size (cm)	Weight (kg)	Sex	Size (cm)	Weight (kg)	Sex
Small	30.5	0.6	F	66.1	5.8	M	71.4	8.0	M
	32.1	0.6	I	71.0	6.8	F	79.2	12.2	F
	34.8	0.7	I	75.3	8.7	M	84.4	15.9	F
	38.9	1.1	I	84.9	12.1	F	87.8	13.2	M
	40.2	1.4	F	87.8	13.3	M	94.2	19.8	M
	42.1	1.4	I	91.9	15.3	M	97.8	22.2	F
Medium	45.5	1.8	M	103.8	25.9	I	102.8	23.8	F
	46.7	2.1	M	109.3	28.3	F	109.5	29.9	M
	49.4	2.6	F	110.1	24.8	F	111.3	30.9	F
	51.0	2.7	F	116.8	32.3	F	115.7	36.6	M
	52.0	3.0	F	127.2	38.5	F	127.7	47.6	F
	55.5	3.9	M	131.3	50.4	M	132.5	51.7	F
Large	56.7	4.1	F	140.4	59.2	M	138.8	63.3	M
	59.2	4.7	M	145.2	58.9	F	142.2	68.5	M
	60.8	5.2	M	151.6	68.7	M	152.8	78.4	F
	63.5	7.7	M	158.8	81.9	M	155.0	90.0	F
	65.5	6.0	F	161.6	89.9	M	162.1	87.3	F
	67.5	6.0	F	164.3	71.0	M	166.8	99.9	M

**Table 2 microorganisms-09-01157-t002:** Results of permutational ANOVAS (PERMANOVA, 999 permutations) performed on Bray-Curtis dissimilarity matrices to test the variation of bacterial community composition with respect to the size of the three tuna species. Bold values indicate a significant effect of the tested factor (*p* < 0.05).

	SKIPJACK				YELLOWFIN				BIGEYE			
	df	Sum of Squares	F. Model	*p* Value	df	Sum of Squares	F. Model	*p* Value	df	Sum of Squares	F. Model	*p* Value
**SIZE**	1	0.25	0.70	0.68	1	0.77	2.02	**0.01**	1	0.48	1.76	**0.04**
**Residuals**	16	5.7			16	6.11			16	4.48		
**Total**	17	5.97			17	6.89			17	4.96		

**Table 3 microorganisms-09-01157-t003:** Bacterial genera representative of the ‘Core Microbiota’ (determined with the microbiome package in R) in the gut of Skipjack, Yellowfin and Bigeye tunas, in the small (S), medium (M) and large (L) size categories. Taxa common to all the three tuna species are represented by grey squares. Red, green and blue squares correspond to unique taxa present in each species.

		SKIPJACK				YELLOWFIN				BIGEYE		
Class	Genus Species	S	M	L		S	M	L		S	M	L
*Actinobacteria*	*Corynebacterium* sp.											
	*Cutibacterium* sp.											
	***Kocuria* sp.**											
	*Microbacterium* sp.											
*Alphaproteobacteria*	*Bradyrhizobium* sp.											
	***Brevundimonas* sp.**											
	*Labrenzia* sp.											
	***Novosphingobium* sp.**											
	***Paracoccus* sp.**											
	*Ruegeria* sp.											
*Bacilli*	*Brochothrix thermosphacta*											
	***Carnobacterium*****sp**.											
	*Lactococcus* spp.											
	***Sporosarcina* spp.**											
	*Vagococcus salmoninarum*											
*Bacteroidia*	***Ulvibacter* spp.**											
	*Vibrionimonas magnilacihabitans*											
*Clostridia*	*Clostridium_sensu_stricto_7* spp.											
	*Gottschalkia* spp.											
	*Tissierella* spp.											
*Fusobacteriia*	*Psychrilyobacter* spp.											
*Gammaproteobacteria*	*Acinetobacter* spp.											
	*Acinetobacter haemolyticus*											
	*Aeromonas* sp.											
	*BD1-7_clade* spp.											
	***Enhydrobacter aerosaccus***											
	*Escherichia/Shigella* sp.											
	*Massilia* sp.											
	*Massilia timonae*											
	***Oceanisphaera* spp.**											
	*Oceanisphaera ostreae*											
	*Photobacterium* spp.											
	*Photobacterium angustum*											
	*Photobacterium leiognathi*											
	***Pseudomonas* spp.**											
	***Psychrobacter* spp.**											
	*Psychrobacter fozii*											
	***Psychrobacter maritimus***											
	*Rahnella* spp.											
	***Shewanella* sp.**											
*Mollicutes*	*Mycoplasma* sp.											

## Data Availability

Not applicable.

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
