# Peer review of "Does the Composition of the Gut Bacteriome Change during the Growth of Tuna?"

_microorganisms, 2021, doi:10.3390/microorganisms9061157_

Round 1

Reviewer 1 Report

Interesting manuscript on the composition of the gut microbiome change during growth of tuna.

The manuscript is well written and the topic is of interest. However, the entire discussion is speculation as the only factor measured is the size of the tuna. There is no measurement  of metabolism, diet or thermoregulation factors related to the tuna which may be clearly related to the microbiome. In addition tuna were fished from May to December and we do not know when each fish was caught. So even within in each size group of each tuna, there might be already diffences to season (and therefore to thermoregulation, diet, etc.). Therefore, this manuscript should not be published as full original paper.

Anyways, the data are important and should be presented in a journal to the scientific community, therefore I would suggest to change this manuscript into a short communication/case report.

Author Response

As reviewer#1 points out, we recognize that our study lacks values for metabolism, diet or thermoregulatory factor, and more generally for all the biotic or abiotic factors that could affect the composition of the gut microbiota. Regrettably, our sampling did not allow us to obtain such data and quantifying all these parameters would in any case have required a colossal and unachievable analytical effort for us. Our main objective was to examine if there was a link between the increase in fish size and the composition of the digestive microflora.578 / 5000

And our data clearly shows that. Without drawing any firm conclusion, we have nevertheless chosen to propose some hypotheses likely to guide the reader in understanding our results; these seem reasonable to us in the light of to the literature on that particular topic.

That being said, given the absence of supplementary data and the possible seasonal bias of our results, we fully understand the remarks of R # 1, and therefore we have decided to significantly reduce the discussion on the possible intervention of these explanatory factors.

The entire paragraph 4.3 « Role of Metabolism » has been deleted, as well as the last paragraph of section 4.2 « Dietary evolution during growth ». We have also removed the sentence from the abstract "Several phylogenic, dietary, and metabolic hypothesis are discussed here". In total, the discussion was reduced by 18.4% (from 1692 to 1380 words), and references 46 to 52 were also removed.

However, we strongly believe that converting this article into a short note would clearly diminish the value of this work and undermine the scientific understanding of our results and of the possible phenomena that could explain the changes in the enteric microflora during the growth of tuna. Before having to substantially modify the format of this paper, we prefer to leave it to the editor to make this choice first.

Reviewer 2 Report

I thoroughly enjoyed reading the manuscript "Does the composition of the gut bacteriome change during the growth of tuna?". The manuscript is extremely interesting, comprehensive, and, barring some typos and difficult to parse sentences, very well-written. However, I think there are some areas which can be improved. 

  • The sentence in the abstract 18-20 "However, during the growth ... skipjack tuna" is extremely hard to parse. Given that this is where you state the central conclusion of your manuscript I would suggest revision.
  • The study concerns the analysis of the microbiome of tuna with respect to two measured variables: species and size. The choice of species as a likely source of variation is obvious, but I would like to see more information in the introduction about why the effect of size on the microbiome is a) likely a priori and b) of interest to the reader. Since the number of potential variables that can affect the microbiome significantly is enormous, it would be good to know if there is, for example, other literature to suggest size-based microbiome variation is important or particularly interesting. Because so many microbiome studies involve collecting as much metadata as possible and then figuring out what is significant afterward, it may be worth handholding the reader a little bit here to explain why no other variables were considered (also relevant to the Statistics section of the methods).

  • In several points in the manuscript, the word "evolution" is used colloquially to mean "change e.g. "the life traits of fish may evolve" in line 39. Given that the word "evolution" has a very specific scientific meaning in the context of the microbiome, I would suggest changing this. 

  • Line 64: what is a "purse seine" fleet?

  • The methods describe that the fish were caught off the African coast and kept on the boat before dissection and fixation at -80 in Abidjan. What was the approximate length of time between catch and dissection, and was it comparable for each sampled fish? The microbiome content of samples can change substantially between sampling and sequencing - I note this in particular reference to the  presence of bacteria found in spoiled meat in the Discussion.

  • Line 79-82 "Considering ... young adults": I would discuss the small, medium and large fish in size order (either ascending or descending) rather than large then small then medium. On the first read I thought it suggested that fish grew non-linearly which was quite a shock. 

  • Line 219-221 "This is contradictory ... zebra fish": this is the sort of information I think would be valuable in the introduction as a justification for why you chose size as the studied variable. 

  • Regarding dietary evolution: this may be a criticism of gut microbiome studies in general, but how much is the microbiome of the gut influenced by the microbiome of the ingested plant / animal matter? If this is just considered a general hazard of studying the gut microbiome, feel free to ignore this. 

Overall I thought the manuscript was good, and with revision would be extremely suitable for a journal like Microorganisms.

Author Response

We thank R#2 for her/his very positive comments.

We agree that the sentence in the abstract was a bit confusing. We have revised it as follows : Although there was a remarkable convergence of taxonomic richness and bacterial composition between yellowfin and bigeyes tuna, the gut bacteriome of skipjack tuna was distinct from the other two species. Throughout fish growth, the enteric bacteriome of yellowfin and bigeyes also showed significant modifications, while that of skipjack tuna remained relatively homogeneous.

To better justifity the choice of size, the following sentence has been added in the introduction : « Knowing that the ontogenic changes typical of certain tuna species allow them to access additional sources of food during their growth [21], it is possible that the diversification of their diet with age (and therefore with size) could result in substantial modifications of their gut bacteriome »

We agree that the word « evolution » can be misleading. It has been replaced by « changes» or « modifications » wherever it was displayed in the document.

A purse seine is a common type of seine used to catch aggregated pelagic species of all sizes from small sardines to the large tunas (http://www.fao.org/fishery/geartype/249/en)

The length time between catch and dissection was, on average, 40 days and it was comparable for each fish. This has been precised in the Material and method « Extraction of the gut bacteriome »

The sentence (Line 79-82 "Considering ... young adults") has been re-organized so that the discussion follows the ascending gradient of fish size (from small to large individuals)

As explained above, we modified the introduction to better justify the reasons for which we examined the effect of size on the gut microbiome. The following sentence has been added in the introduction : « Knowing that the ontogenic changes typical of certain tuna species allow them to access additional sources of food during their growth [21], it is possible that the diversification of their diet with age (and therefore with size) could result in substantial modifications of their gut bacteriome »

The question about whether the gut microbiome could be influenced by the microbiome of the ingested plant/animal is very interesting but also very difficult to quantify. In humans, we have long known about the role the microbiome of consumed animals can play in triggering food poisoning. However, there are very few studies to our knowledge that have adressed these questions in marine organisms which would undoubtedly merit further study.

Reviewer 3 Report

This is an interesting and nice paper that provide gut bacteriome grown in tuna species. Although similar study had been reported during growth of fish (reference 18 and 19) but present study found interesting.

Although paper written clearly but need to address following to improve manuscript

Authors clearly present growth of bacteriome in different tuna species, However, the mechanism by which these growth factors effect need to be discussed in thoroughly.

I would also suggest revising the tittle of the manuscript in a specific

does chilled brine had any effect on sampling process?

Out of 15 different class identified, is any effect of marine environmental and dietary source effect the grown and development of fish and gut microbiome this should be validated and  discussed.

Authors observed microbiological difference observed with the same species, how this gut help to maintain healthy host should be presented in results.

Author Response

The first comment is contrary to R#1 who found that these factors should not be discussed since they are not supported by any data/measurements. R#1 even suggests to reduce this specific part of the discussion.

We are sorry we do not understand the comment about the title and how we should revise it

The question about the possible incidence of chilled brine is interesting, however, regrettably, we did not address it. However, if the freezing treatment on the fish with chilled brine could potentially alter the composition of their skin microbiome, there is less reason to think that this would be also the case with the gut microbiota which is not in direct contact with the freezing agent., since the intestinal tract is not in direct contact with. Also the effect of this brine could be questionable if the composition of the gut bacteriome was homogeneous between the different samples; here on the contrary, there are strong differences between tuna species and size classes.

Since the gut content (as proxy of diet) of the fish has not been investigated here, nor the environmental conditions at each fishing sites, we cannot further discuss these factors, as also suggested by R#1

Round 2

Reviewer 1 Report

The authors did an excellent job in rewriting the discussion without using any speculations. I have no further objections, no further revisions are necessary.  The way it is now presented I agree with the authors to publish it as a original paper.